# The Time-Varying Effect of Participatory Shift Scheduling on Working Hour Characteristics and Sickness Absence: Evidence from a Quasi-Experiment in Hospitals

**DOI:** 10.3390/ijerph192214654

**Published:** 2022-11-08

**Authors:** Jarno Turunen, Kati Karhula, Annina Ropponen, Aki Koskinen, Rahman Shiri, Mikael Sallinen, Jenni Ervasti, Jaakko Pehkonen, Mikko Härmä

**Affiliations:** 1Finnish Institute of Occupational Health, P.O. Box 40, FI-00032 Helsinki, Finland; 2Jyväskylä University School of Business and Economics, FI-40014 Jyväskylä, Finland; 3Division of Insurance Medicine, Department of Clinical Neuroscience, Karolinska Institutet, 171 77 Stockholm, Sweden

**Keywords:** self-rostering, shift schedule, sickness absence, working hours

## Abstract

Participatory shift scheduling for irregular working hours can influence shift schedules and sickness absence. We investigated the effects of using participatory shift scheduling and shift schedule evaluation tools on working hour characteristics and sickness absence. We utilized a panel data for 2015−2019 with 16,557 hospital employees (6143 in the intervention and 10,345 in the control group). Difference-in-differences regression with ward-level clustered standard errors was used to estimate the average treatment effect on the treated coefficients relative to timing of the intervention with 95% confidence intervals (CI). Using participatory scheduling tool increased long working hours and weekend work and had delayed effects on the short (1–3 days) sickness absences. Increased effects were observed: 0.2 [95% CI 0.0−0.4] days for the second, and 0.8 [95% CI 0.5−1.0] for the third year after the onset of intervention. An average increase of 0.5 [95% CI 0.1−0.9] episodes on all sickness absence episodes was observed for the third year. Using the shift schedule evaluation tool with the participatory shift scheduling tool attenuated the adverse effects. To conclude, participatory shift scheduling increased some potentially harmful working hour characteristics but its effects on sickness absence were negligible, and further attenuated by using the shift schedule evaluation tool.

## 1. Introduction

Irregular working hours including evening and night shifts are required in many occupations. Hospitals covering specialized inpatient health services operate 24/7, 365 days a year. This imposes a demand for hospitals to organize irregular working hours in a way that considers the related health and safety issues, in addition to operational needs. At the working population level, meeting this demand is also of importance, since in the EU28-countries, shift work was most common in the healthcare sector with 40% of employees working in shifts [1].

The literature reviews on shift work have indicated associations with insufficient sleep and accidents [2], chronic diseases and conditions [2,3], and poorer mental health [4]. Sickness absence is considered as a relevant measure of ill health and functioning [5], and the related costs to individual employees, employers, and society are substantial [6,7,8,9]. Various working hour characteristics in healthcare such as a high number of consecutive night shifts, long shifts of >12 h, and short shift intervals of <11 h have been shown to be associated with the increased risk of sickness absence [10,11]. Yet, the evidence on long working hours is inconclusive and suggests that the widely existing negative relationship between long work hours and sickness absence may partly be due to the healthy worker effect [12].

Evidence from a systematic review on nurses suggests that the health problems associated with shift work are mediated by at least chronotype and sex [13]. Besides selection or individual preferences, a study from Finnish healthcare found that there is no association between longer working hours and the risk of sickness absence in the short exposure time of 10 to 30 days, and the risk is lower on exposure of 40 days or more [14], suggesting that the exposure time window plays an important role in the potential health effects of long working hours. Other studies on nurses have found that it is not only shift work that affects health, but rather how shift work is organized in terms of weekly hours, irregular schedules, and length of recovery periods [11,15,16,17]. The findings on shift work and health, particularly in the healthcare sector, underline the importance of taking into account the total duration of exposure to shift work when examining the link between shift work, working time characteristics, and sickness absence.

One way to promote employees’ health when working irregular hours is to minimize the occurrence of shift characteristics known to be associated with impaired health and safety [18]. The impaired ergonomics of irregular working hours in shift work, and potentially harmful working hour characteristics pose health and safety risks to employees. To avoid these adverse effects, fewer consecutive night shifts, sufficient recovery periods between shifts, and shorter night shifts, for example, have been reported as ways to optimize sleep and reduce fatigue and related problems such as health impairments or accidents [19]. Regardless of the individual preferences of shift workers, safer and healthier schedules can be achieved by focusing on the ergonomics of the scheduled shifts. Our recent study showed that the use of shift schedule evaluation tools was associated with improvements in many, but unfavorable changes in some working hour characteristics [20]. The findings suggested that at least modest changes toward more ergonomic shift schedules can be achieved by using the shift schedule evaluation tool including recommendations for better shift ergonomics.

Another employee-oriented way of organizing irregular working hours is to enable the employees to have more control over their working hour characteristics (e.g., position, duration, and distribution) [21]. Such control can be achieved by giving employees the means to self-roster shifts. Cross-sectional studies on various occupational populations have shown individual high work time control (WTC) to be associated with a number of improved health and well-being-related outcomes. Refs. [22,23,24,25], suggesting that improved WTC in shift work can improve health and well-being. A recent review on employee-oriented flexible work and mental health suggested that flexible working arrangements increase WTC but have at best modest beneficial effects on mental health [26]. As it stands, the evidence on WTC allows, however, limited possibilities for causal inference [27].

To date, self-rostering interventions aimed at improving WTC have been predominantly studied in healthcare or related fields [28,29,30,31,32,33]. Findings support the potential health and well-being effects of self-rostering to some extent. Among eldercare workers, involvement in the planning of work schedules did not influence health and well-being measures [28]. Another study compared outcomes of three different interventions enhancing WTC, with two of them implemented in healthcare: first, the possibility to adjust starting time and length of shift in 15-min intervals, and second, the possibility to choose working days and predefined duties [29]. Whereas both studied interventions improved WTC, the observed health and recovery improved more in the group where the WTC increased less than in the first group. Moreover, the observed effects could not be explained by changes in working hour characteristics. The same interventions offered positive effects on job demands and the social environment in the workplace [30]. Participatory scheduling as a way of self-scheduling has been shown to increase long work shifts [31], and to reduce short sleep, poorly perceived workability [32], and ward-level sickness absence [33]. However, self-rostering may also lead to impaired working hour ergonomics, as a study on police officers’ shift work preferences has shown [34].

The studies [28,29,30,31,32,33] on self-rostering in shift work offer somewhat mixed evidence. The previous findings indicate that while self-rostering and participatory shift scheduling can offer an improvement in WTC, health, and well-being, irregularity of working hours may increase as a result. The findings also suggest that the increase in the irregularity of working hours (e.g., variability of shift starting and ending times, and the length of the work shifts and spells), may have adverse health effects and consequently lead to increased sickness absence [10,11,14,15,16,17]. On the other hand, the use of shift schedule evaluation tool may improve the ergonomics of shift schedules and thus offer ways to reduce those working hour characteristics in irregular shift work that may have negative effects on health and wellbeing (e.g., long work shifts and quick returns, i.e., evening-morning shift combinations) [20].

The focus of this study is the investigation of the effects of self-rostering on shifts by using the participatory shift scheduling tool alone or in combination with the shift schedule evaluation tool. The participatory scheduling tool offers an opportunity for an individual employee to self-roster into shifts see e.g., [32]. The shift schedule evaluation tool offers feedback in evaluating the ergonomics of shift schedules see e.g., [20]. Together the two shift scheduling tools, the participatory shift scheduling tool and the shift schedule evaluation tool studied here can have both impairing and improving effects on working hour characteristics, irregular hours in shift work, and consequently, both increasing and decreasing effects on sickness absence.

We aimed to investigate the dynamic treatment effects of the use of the participatory scheduling tool alone, and in combination with the shift schedule evaluation tool, on various working hour characteristics, the number of sickness absence days from short (1–3 days) sickness absences, and the number of all sickness absence episodes. Our focus was on short-term sickness absence days, but we also calculated the total number of sickness absence episodes irrespective of the length of the episode. Short-term sickness absences were selected as the focus because they can potentially reflect self-perceived health or motivation to work, i.e., they are not always associated with a disease condition or symptoms of any kind [5]. However, we also included sickness absence episodes irrespective of their length, as the increase of short-sickness absences can potentially be observed as changes in the total number of sickness absence episodes. This study contributes to self-rostering literature by investigating the effect of using the participatory shift scheduling tool on individual hospital employee working hour characteristics and sickness absence. We base our evidence on a cohort study analyzed as a quasi-experiment. We offer a view on the timing of the effects, whether they are immediate or delayed and/or short-lived or long-lasting. Furthermore, we estimate the effect of using the ergonomics-enhancing shift schedule evaluation tool in combination with the participatory shift scheduling tool.

## 2. Materials and Methods

### 2.1. Sample of Participants

This cohort study analyzed as a quasi-experiment is a part of the ongoing Working Hours in the Finnish Public Sector (WHFPS) study using register-based payroll data of working hours described in detail previously [35]. Employer’s register data was used with their permission, and all the data was pseudonymized.

The sample of participants consisted of hospital employees who worked in 9 hospital districts in the Finnish public healthcare sector during 2015−2019. These hospital districts cover specialized inpatient health services around the clock and have used Titania^®^ (CGI Finland Ltd, Helsinki, Finland) shift scheduling software in the scheduling of shift work.

We used a balanced panel of hospital employees that were followed each year from 2015 to 2019. Inclusion criteria were (i) ≥31 shifts each year, and (ii) shift work by having ≥10% evening and/or night shifts of all shifts during the years 2015−2019. In more detail, the data for 2015−2019 included payroll data on the working hours of 135,659 employees in 21 organizations, including 10 hospital districts and 11 municipalities. Generating a balanced panel on the use of different work scheduling tools and sickness absence data with yearly observations for each employee for the years 2015−2019 excluded 108,454 employees who worked in the cities/municipalities but had less than five years of data leading to a total number of 27,205 employees in 9 hospital districts. Excluding employees due to inconsistencies in the data, such as missing working hour characteristics or missing data on sickness absence led to a sample of 25,725 employees. The restriction to those working ≥10% of the evening and/or night shifts led to the final sample of 16,488 employees. They worked in 1337 hospital wards. The total number of individual-year-observations was 82,440.

### 2.2. Shift Characteristics and Sickness Absence Outcomes Measures

Our main outcomes of interest were the number of sickness absence days from short absences of 1–3 days and the number of sickness absence episodes from all sickness absences. Both outcomes were continuous measures of all-cause sickness absence. We had no information on diagnosis-specific or medically certified sickness absence. The studied working hour characteristics were percentages of >40-h working weeks, >48-h working weeks, >12-h shifts, long spells of working shifts, <11-h shift intervals, <28-h recovery periods after last night shift, realized shift plans, weekend work, and single free days, as described earlier [35]. All the measures were based on payroll-based register data aggregated to individual-level annual measures.

### 2.3. Interventions and Identification Strategy

We considered the use of the participatory scheduling tool as the primary treatment investigated in this study. The use of the shift schedule evaluation tool was investigated only in combination with the participatory scheduling tool. The users of these tools formed different intervention groups that are referred to as intervention group, double-intervention group, and control group, respectively.

The decision to use shift scheduling tools was made at the hospital district level. The final decision to use the participatory scheduling tool was done in the wards, and for the use of the shift schedule evaluation tool, the decision to use it, and make changes to the schedule in accordance with the recommendations towards less harmful working hour characteristics was made by individual employees.

The participatory scheduling tool allowed the hospital employees and the head nurse or shift planner to interactively roster the shift schedules. The participatory scheduling is based on agreed ward-level rules and principles. These rules may include, for example, a minimum number of weekend and night shifts, order of inputting shifts, and shift types, for example, night, evening, and day shifts, into the software. In short, each employee enters one’s own desired shifts into the wards’ shift plan according to the agreed rules, the operational demands forward, and other employees’ inputs. Eventually, participatory scheduling is an obligation and not only an opportunity to participate in the rostering of shift schedules. The head nurse or shift planner is responsible for the initialization of the work scheduling process, checking, and accepting the final shift schedules. Participatory shift scheduling tools introduce an interactive way for hospital employees to schedule their shifts collaboratively, taking into account the ward-level operation, working time legislation, and fairness and equality for the employees [36].

The shift schedule evaluation tool offers feedback and helps in evaluating the work schedules according to timing and length of working hours, a number of consecutive night shifts, and recovery time between shifts. In the standard version of the software, the evaluation tool can be used only by the shift planner of the ward. In the participatory shift scheduling tool, the shift schedule evaluation tool can be used by both the shift planner and the employees [20]. The schedule evaluation tool can be used to detect the potential health and well-being associated drawbacks in the working hour schedule, and to revise the schedule if desired. As opposed to the use of the participatory scheduling tool, the use of the shift schedule evaluation tool by individual employees was voluntary.

We utilized data from multiple time periods, 2015−2019, and variations in intervention timing. Eventually, 1847, 2272 and 2024 employees started the use of the participatory scheduling tool in 2017, 2018, and 2019, respectively. Throughout the years 2017−2019, there were 849 employees who dropped out from the intervention i.e., switched to a ward that did not use the participatory scheduling tool. The dropouts were added to the control group. The control group was a combination of never-treated 10,345 hospital employees, i.e., those that did not receive intervention at all, and a time-varying group of those who were not-yet-treated, i.e., received the intervention later. For example, during 2015 and 2016 no one received the intervention, during 2017 the first 1847 belonged to the intervention group, and all the rest formed the control group. In 2018 there were 4119 employees in the intervention group, and eventually, in 2019 there were 6143 employees in the intervention group and 10,345 in the control group. The use of the shift schedule evaluation tool in combination with the participatory scheduling tool was started by 431, 723, and 860 employees in 2017, 2018, and 2019, respectively. The control group was a combination of never-treated and those not-yet-treated.

Our identification strategy is based on the inclusive payroll data information regarding the use of Titania^®^ shift scheduling software tools. We could obtain every single 3-week period during 2015−2019 for which the employees used (or did not use) the participatory scheduling and the shift schedule evaluation tools. This offered us detailed measures for the intervention and control conditions. The participatory scheduling intervention was considered ongoing when 80% of employees’ annual 3-week periods had been planned using the participatory scheduling tool. For the shift schedule evaluation tool, the measure for the treatment to be in effect was the actual checking and change made by the employees to the shift schedule in accordance with the shift schedule evaluation tool. The first use was considered as the intervention since the ergonomics of the shift schedules can be both learned and checked rather easily via the shift schedule evaluation tool. Both interventions are considered absorbing: once they have been started, they remain in use.

### 2.4. Statistical Analyses

Research designs applying data with multiple periods and variation in intervention timing usually include both individual fixed effects and time fixed effects in regression specification to estimate the treatment effect parameter coefficients while controlling for unobserved individual characteristics, and unobserved time effects that are common for all individuals in the sample [37]. The resulting estimator is called a two-way fixed effects estimator and it is considered a generalization of the difference-in-differences estimator. We employed a doubly robust two-way fixed effects estimator [37,38] to estimate the treatment effect parameter coefficients with 95% confidence intervals and ward-level clustered standard errors. We extended this difference-in-differences estimator to the event-study framework to study treatment effects relative to the timing of the treatment in a six-year time window. We focused on reporting the estimates for three years prior to the intervention, and three years after the onset of the intervention. The estimated event study regression specification was:$$Y_{it}=\alpha_{i}+\lambda_{t}+\overset{T-1}{\underset{l=-T}{\sum }}D_{it}^{l}{µ}_{l}+\beta^{\prime }X_{it}+u_{it}$$
where the outcome of interest $Y_{it}$ is the number of sickness absence days from short absences of 1–3 days, sickness absence episodes, or percentage of working hour characteristics of employee *i* during year *t*. ${µ}_{l}$ is the effect on individual-level outcome relative to intervention (or double intervention) implemented at time 0 in ward *i* during year *t*, and *l* describes the number of periods that individual employee has been exposed to the intervention. $\alpha_{i}$ and $\lambda_{t}$ are individual and time dummies for each individual *i* and year *t*, respectively. *β* is a vector for the coefficients and $X_{it}$ is a vector for the time-varying independent variables, such as age, the nursing profession, ward size, the share of evening or night shifts of all shifts, and working hour characteristics, including percentages of >40-h working weeks, >48-h working weeks, >12-h shifts, long spells of working shifts, <11-h shift intervals, <28-h recovery periods after last night shift, realized shift plans, weekend work, and single free days. $u_{it}$ represents standard errors clustered on the ward level.

The specification (i) allowed the estimation of regression coefficients relative to the timing of the intervention, offering a view to the timing of the effects i.e., whether they are immediate or delayed, short or long-lasting, (ii) improved the credibility of the findings by investigating potential pre-intervention effects in detail, and (iii) estimated the regression coefficients using a combination of outcome regression modeling and inverse probability weighting. The baseline measures from the year 2015 for the variables included in the regression specification were used to calculate inverse probability weights of belonging to the intervention group. For this so-called doubly robustness of estimates it suffices that either the outcome regression or the propensity score model is correctly specified. In other words, conditional parallel trends between the treatment and control groups needed to exist for the outcome regression model, or the correct modeling of the conditional probability of individual hospital employee i being in intervention group g given covariates X was needed for the inverse probability weighting. The key assumption for our estimators to be valid was that employees with the same characteristics would follow the same outcome trend in the absence of the intervention. The second assumption needed was that there is no anticipation behavior related to intervention i.e., individuals do not adjust their sickness absence behavior before the intervention.

The treatment effect parameter coefficients were estimated as difference-in-differences between the intervention (or double-intervention) and control group, relative to the timing of the intervention. For example, the coefficient for two years before the intervention was the difference in outcomes between the intervention and control groups obtained in the following way: for those who started the intervention in 2017 (or 2018, 2019, respectively), the outcome measure two years before the intervention, i.e., in the year 2015 (or 2016, 2017, respectively) was compared to the outcome of the control group in 2015 (2016, 2017, respectively). The average of all these comparisons is the average treatment effect on the treated two years before the intervention. A similar comparison is a basis for all the estimated dynamic treatment effects in a time window of three years before the intervention and three years after the onset of intervention.

We controlled for various measures in the estimation of treatment effects for working hour characteristics outcomes: age, nursing profession (e.g., registered nurse, nursing assistant, or midwife), sex, ward size, and the amount of shift work i.e., share of the evening and night shifts. These variables were also used in inverse probability weighting for the doubly robust estimator. In the estimation of treatment effects for sickness absence outcomes we controlled for age, nursing profession, sex, ward size, the amount of shift work, and various working hour characteristics related to the length of working hours, shift intensity, and social aspects of shift work such as weekend work and single free days. These variables were also used in inverse probability weighting for the doubly robust estimator. The two-way fixed effects estimator also included individual fixed effects and time-fixed effects. In other words, we controlled for unobserved individual effects that do not change over time and unobserved time effects that are common to all individuals in the sample. Such unobserved individual fixed effects could be e.g., chronotype, or a long-lasting health condition. An example of a common time effect is, for example, the macroeconomic situation, which is known to be associated with the sickness absence of women working in the public sector [39].

We tested the conditional parallel trends assumption by using the event study regression. We aimed to show that there are no pre-intervention difference-in-differences in the sickness absence outcome measures between the intervention and control groups. This was done to find support for our main identification assumption that employees with the same characteristics would follow the same outcome trend in the absence of the intervention.

We carried out robustness check in addition to the main analysis. We estimated the average treatment effect for different timing groups, i.e., for groups that started the intervention(s) in 2017, 2018, and 2019 to explore potential starting group effects in more detail.

## 3. Results

### 3.1. Descriptive Statistics in the Baseline Year 2015

The baseline characteristics for the intervention and control groups are reported in Table 1, left panel, separately for the use of the participatory scheduling tool in combination with the shift schedule evaluation tool i.e., double intervention and the control group (right panel).

Mean age for the intervention group was 42 years and for the control group of never-treated 43 years at the baseline in 2015 (Table 1, left panel). The share of women was higher in the intervention group (88%) than in the control group (84%). There were more employees working in nursing professions such as a nurse, midwives, and assistant nurses, in the intervention group (85%) in comparison to the control group (64%). The hospital employees in the intervention group worked in wards with an average number of 30 employees, while employees of the control group worked in wards with the average number of 27 employees.

The intervention group had more night shifts of all shifts. There was neither difference in the share of evening shifts of all shifts between the groups, nor in the share of >40-h working weeks. The employees in the intervention group worked fewer >48-h working weeks in comparison to the control group. The >12-h shifts were more common in the control group than in the intervention group. The intervention group had fewer long spells of work shifts and <28-h recovery periods after the last night shift. The intervention group had more <11-h shift intervals, realized shift plans, weekend work, and single free days.

There were no statistically significant differences in sickness absence variables at the baseline between the intervention and control groups (Table 1, left panel). Both groups had on average 4 short sickness absence days and 12 days from all sickness absences and on average 2 episodes of short sickness absences, and 3 episodes of all sickness absences.

Regarding the combined use of participatory scheduling and shift schedule evaluation tools, there were more women in the double-intervention group in comparison to the control group (89 vs. 85%, respectively) (Table 1, right panel). The employees in the double-intervention group worked more often in the nursing profession (86 vs. 70%) and were on average younger in comparison to the control group (40 vs. 43 years). The average number of personnel in the wards of the intervention group had 29 employees whereas in the wards of the control group there were 28 employees.

There was no difference in the share of evening shifts between the groups, but the double-intervention group did more night shifts of all shifts. The double-intervention group had more >40-h work weeks but less >48-h working weeks, and >12-h shifts. Both groups had approximately 4% of long spells of work shifts of all spells of shifts. The double-intervention group had considerably more <11-h shift intervals, and less <28-h recovery periods after the last night shift. The double-intervention group had more weekend work and more single-free days. There was no statistically significant difference in the realized shift plans between the groups.

There were no differences in short or all sickness absence days between the double-intervention and control groups. However, there were minimal differences in the short and all episodes of sickness absence between the groups.

### 3.2. Dynamic Treatment Effects on Working Hour Characteristic Measures

The average treatment effects on the treated (ATT) on the working hour characteristics conditional on the covariates are presented in Table 2. The treatment effect parameter coefficients were estimated changes in percentage points of the occurrence of working hour characteristics. Age, sex, nursing profession, ward size, and the share of evening and night shifts were controlled in the regression specification. The first specification estimated the effects of the use of the participatory scheduling tool. The second specification estimated the use of both the participatory scheduling tool and the ergonomics evaluation tool in combination. The statistical significance is depicted by 95% confidence intervals excluding zeroes.

There were pre-existing differences between the intervention and the control group in >40-h working weeks, >48-h working weeks, long spells of work shifts, realized shift plans, weekend work, and single free days (Table 2). There were no pre-existing differences between the intervention and the control group in <11-h shift intervals nor <28-h recovery periods after the last night shift.

The observed pre-intervention differences were rather negligible. For example, the observed difference of −0.5 pp [95% CI: −1.0; −0.0] suggests that the intervention group had one working week of >40-h less compared to the control group two years prior to the intervention. Similarly, the observed difference of 0.3 pp [95% CI: 0.2; 0.5] in long spells of work shifts suggests that there was a difference of less than one long spell of work shifts one year prior to treatment. The observed difference in weekend work, 1.0 pp [95% CI: 0.3–1.7], suggests that the intervention group worked one to two more weekends three years prior to the intervention.

The occurrence of >40-h working weeks increased by 0.9 pp [95% CI: 0.5−1.3] during the first full year of the intervention (Table 2). For the second year, there was an effect of 0.6 pp [95% CI: 0.1; 1.2]. The longer >48-h working weeks increased by 0.6 pp [95% CI: 0.4; 0.8] during the first full year of intervention, and by 0.5 pp [95% CI: 0.2; 0.9], and 0.8 pp [95% CI: 0.3; 1.3] during the second and third year after the intervention, respectively.

Long >12-h shifts increased by 0.4 pp [95% CI: 0.1; 0.6] during the year of the intervention, and by 0.8 pp [95% CI: 0.1; 1.5] for the third year. There was no statistically significant difference between the intervention and control groups regarding the changes in percentages of <11-h shift intervals. Shorter <28-h recovery periods increased in the third year after the intervention by 1.0 pp [95% CI: 0.1; 1.9].

There was no difference in the realization of shift plans after the use of the participatory scheduling tool. Weekend work increased after the onset of the intervention, by 1.8 pp [95% CI: 1.5; 2.2] during the year of the intervention, and by 2.6 pp [95% CI: 2.0; 3.1], and 4.0 pp [95% CI: 3.1; 4.9] during the second and third year, respectively. Single free days increased during the first, second and third year of the intervention by 0.4 pp [95% CI: 0.2; 0.6], 0.7 pp [95% CI: 0.3; 1.0], and 0.8 pp [95% CI: 0.1; 1.2], respectively.

The use of the shift evaluation tool in combination with the participatory scheduling tool had somewhat analogous pre- and post-intervention effects in comparison to the use of the participatory scheduling tool only (Table 2). There were no pre-existing differences in the percentages of >40-h working weeks, >48-h work weeks, >12-h shifts, <28-h recovery periods after the last night shift, nor in the realization of shift plans. There was an observed pre-intervention difference in the long spells of work shifts of 0.2 pp [95% CI: 0.0; 0.4]. Weekend work was more common in the double-intervention group by 1.3 pp [95% CI: 0.4; 2.1], 0.9 pp [95% CI: 0.3; 1.5], and 1.1 pp [95% CI: 0.6; 1.6] three, two and one years before the intervention, respectively. Single-free days were more common in the double-intervention group two years prior to intervention by 0.7 pp [95% CI: 0.3; 1.0].

During the first full year of the double-intervention there were increases in >40-h working weeks by 0.9 pp [95% CI: 0.4; 1.4], >48-h working weeks by 0.5 pp [95% CI: 0.2; 0.8], weekend work by 1.3 pp [95% CI: 0.8; 1.8], and single free days by 0.4 pp [95% CI: 0.1; 0.7]. During the third year, there was a difference of 0.9 pp [95% CI: 0.1; 1.6] in the >48-h working weeks. All the other differences relative to intervention timing were statistically non-significant.

### 3.3. Dynamic Treatment Effects on Sickness Absence Outcome Measures

The average treatment effects on the treated on the sickness absence measures are presented in Table 3 and Figure 1, Figure 2, Figure 3 and Figure 4. The treatment effect parameter coefficients are estimated changes on the days and episodes of sickness absence, respectively. Age, sex, nursing profession, ward size, and the share of evening and night shifts, and the working hour characteristics including the length of working hours, shift intensity, and social aspects of shift work were controlled in the regression specification. The first specification estimates the effects of the use of the participatory scheduling tool. The second specification estimates the use of both the participatory scheduling tool and the ergonomics evaluation tool in combination.

For both dependent variables, there was full support for the conditional parallel trends assumption to hold. There were no differences in pre-intervention outcomes for the intervention and control groups (Table 3).

The short sickness absences increased by 0.2 days [95% CI: 0.0; 0.4] and 0.8 days [95% CI: 0.5; 1.0] during the second and third year after the onset of intervention, respectively (Table 3 and Figure 1). These estimates correspond to increases of 6% and 22% in comparison to the baseline year 2015. The effect estimated for the third year is based on those who started the use of the participatory scheduling tool in 2017. The number of sickness absence episodes remained rather stable between the intervention and control group before the treatment, and during the first two years after the onset of intervention (Table 3 and Figure 2). The estimated effect for the third year was 0.5 episodes [95% CI: 0.4; 0.7], or a 20% increase in comparison to the baseline year. Again, the effect estimated for the third year is based on those who started the use of the participatory scheduling tool in 2017.

In the second specification we estimated the treatment effects of using the participatory scheduling tool in combination with the shift schedule evaluation tool. The estimated effect on the short sickness absence days was smaller in magnitude and remained non-significant before the third year after the onset of intervention (Table 3 and Figure 3). The estimated effect for the third year, based on those that received the intervention in 2017 only, was an increase of 0.5 days [95% CI: 0.1; 0.9], or a 15% increase when compared to the baseline year 2015. The number of sickness absence episodes shared the same type of course. The estimated effects were close to zero before the intervention and for the first two years after the onset of the intervention (Table 3 and Figure 4). The third-year estimated effects based on only those who received double intervention in 2017 was 0.3 episodes [95% CI: 0.1; 0.6], or a 12% increase when compared to the baseline year 2015.

### 3.4. Robustness Analysis

The robustness analysis includes the average treatment effects for all the treated, for all treatment years (Appendix A, Table A1). For the use of the participatory scheduling tool only, the estimated average treatment effect on the treated was 0.2 days [95% CI: 0.1; 0.4] increase on the short sickness absence days. Further analysis showed that the estimated effect was larger for those who started the intervention in 2017 [ATT: 0.3 with 95% CI: 0.1; 0.5] in comparison to those who received the intervention later in 2018 [ATT: 0.2, 95% CI: 0.0; 0.4], or in 2019 [ATT: 0.2, 95% CI: −0.1; 0.4].

The estimated treatment effect for the sickness absence episodes was an increase of 0.1 episodes [95% CI: 0.0; 0.2]. All the different intervention timing groups, whether started in 2017, 2018, or 2019, had the same estimated effect of 0.1 episodes. However, the effects of those that received the intervention in 2018 or 2019 were statistically non-significant.

For the use of the participatory scheduling tool in combination with the shift schedule evaluation, all the estimated average treatment effects on the treated were smaller in magnitude in comparison to the estimated average treatment effects on the treated with the use of the participatory scheduling tool only. The estimated effects were also non-significant.

## 4. Discussion

In this cohort study analyzed as a quasi-experiment we aimed to investigate the dynamic treatment effects of using the participatory scheduling tool alone, and in combination with the shift schedule evaluation tool, on the occurrence of various working hour characteristics, on the number of sickness absence days from short absences of 1–3 days, and on the number of all sickness absence episodes. We compared the difference-in-differences between individuals who started using the participatory shift scheduling tool and the shift evaluation tool to those who did not.

We found that using the participatory scheduling tool enables more work-time control to have immediate but minor effects on working hour characteristics that may potentially have negative effects on health and wellbeing. In comparison to baseline measures from the year 2015, these differences can be considered rather negligible. We found a delayed negative effect of using participatory scheduling tools on short sickness absence days and all sickness absence episodes. For the third year of use, we observed a 22% increase in short sickness absence days and a 20% increase in all sickness absence episodes. The finding on the delayed third-year effect was solely based on the employees who started using the participatory scheduling tool in 2017. The average treatment effect on the treated estimated for the whole intervention time pointed to an increase of 6% for the short sickness absence days, and 4% for the episodes, respectively. All the estimated coefficients on working hour characteristics and both sickness absence outcomes were attenuated by using the shift schedule evaluation tool in combination with the participatory scheduling tool.

The intervention and control groups were similar regarding various baseline measures. Both intervention groups had more employees working in nursing professions such as nurses, midwives, and nursing assistants. The double-intervention group, i.e., those that used both the participatory scheduling tool and the shift schedule evaluation tool, was younger and healthier in terms of sickness absence measures in comparison to the control group. We took into account these differences between the groups in our regression specifications, both the regression outcome regression modeling and the inverse probability weighting based on baseline values. We also included the not-yet-treated in the control group. The existence of conditional parallel trends and no anticipation behavior would need to meet in support of our findings. We did not find signs of pre-intervention differences in the sickness absence outcomes, although we did find some negligible differences in pre-intervention working hour characteristics. We found support for the assumption of conditional parallel trends. It is also reasonable to assume that there are no anticipation effects regarding sickness absence outcomes before the start of using the shift scheduling tools. Thus, the estimates offered here can be considered valid measures of using participatory scheduling tools alone, or in combination with shift schedule evaluation tools on sickness absence and various working hour characteristics.

The use of the participatory scheduling tool increased working hour characteristics such as long working weeks, >12-h shifts, weekend work, and single free days that may have negative effects on health and wellbeing, although we also observed minor pre-treatment effects (Table 2). This is in line with effects previously found in other studies [31,33] and may reflect the preference of the employees to obtain longer off-time periods by preferring compressed working hours reflected as longer work shifts and/or longer work spells [40]. Since the collective agreements did not allow planning shifts with overtime, and the average working hours were to be balanced every 3 weeks, the use of long working hours support in practice longer times off. For the most part, these potentially negative effects of shift ergonomics associated with compressed working hours were minor in magnitude, immediate, yet long-lived as they existed for the follow-up period after the onset of treatment.

The changes following the use of the participatory scheduling tool in working hour characteristics in comparison to baseline measures in the year 2015 can be considered rather negligible. However, these observed effects were attenuated with the use of the shift schedule evaluation tool in combination with the participatory scheduling tool. This finding is in line with our previous study on the effects of the shift schedule evaluation tool used by the head nurse or shift planner on working hour characteristics [20]. Consequently, one could expect associated improving health effects, as was shown in the current study, as the magnitude of estimated effect coefficients on sickness absence were small for those in the double-intervention group. This suggests that while the increased self-rostering leads to occasionally long working hours and compressed weeks, the use of the shift schedule evaluation tool supported the evaluation of the risk and led to the modification of the work schedules to decrease the potential hazards of the long working hours and smaller increases in sickness absence throughout the years.

The overall findings of the effect of participatory shift scheduling on sickness absence are not in accordance with our previous results that reported a decrease in ward-level sickness absence after using the participatory scheduling tool [32]. There are at least a few reasons that can explain this. First, building a balanced panel of five years for hospital employees ends up in a selected group that consists of employees that had enough work shifts, and continued shift work during the five-year follow-up. Second, our previous study is based on using ward-level average sickness absence, which is a result of not only the numerator but also the denominator. The third reason for the different results could be employee turnover in wards, which is removed when building a balanced panel data. Fourth, increased sickness absence may introduce a demand for more employees in the ward, introducing smaller average sickness absence. Individual level effects, dynamic treatment effects, and potentially heterogenous effects of self-rostering merit future investigation.

Our study used objective, large-scale register data of hospital employees. The working hour characteristics, sickness absence, background information as well as the two intervention measures used were based on objective payroll-based hospital records. We defined the interventions conservatively, 80% threshold for annual participatorily scheduled periods, and changes made to the shift schedule after the use of the shift schedule evaluation tool. By using the event study regression specification, we were able to estimate the time-varying effect on outcomes. We showed evidence of the conditional parallel trends assumption in support of our estimated results. Our estimates were doubly-robust, i.e., it suffices that either the outcome regression for outcome variables or the inverse probability weighting is correctly specified. We were able to show whether the effects are immediate, delayed, short-lived, or long-lasting. With the use of balanced panel data, we were able to control for unmeasured individual-level confounders that do not change in time, such as chronotype. Moreover, we were able to control for time-varying shocks which are common for all employees, such as macroeconomic situations. However, the unmeasured time-varying individual-level factors, such as lifestyle factors, cannot be statistically controlled in our regression analyses.

To the best of our knowledge, this study is among the first studies to estimate the effect of self-rostering of any kind on individual-level sickness absence and estimate the time-varying effects. In addition, we combined self-rostering carried using participatory scheduling with the use of a shift schedule evaluation tool enhancing shift ergonomics. These findings show that there is more to consider than simple pre-post estimates exploiting short follow-up periods when investigating the effects of different interventions aimed at healthcare staff. Health condition(s), and sickness absence as a measure for those, take a long time or long exposure to appear. Our results show that the effects of participatory scheduling on sickness absence are delayed. By focusing on shorter follow-ups, these results would not have been obtained. This is in accordance with the findings regarding longer working hours and sickness absence suggesting taking into account the exposure time window [14]. Our results indicate that prolonged exposure to potentially hazardous working hour characteristics may have negative effects on health and wellbeing, which in turn may influence the delayed increase in sickness absence measures. However, the findings should be interpreted with caution as the differences in working hour characteristics are minor.

Our study has several limitations. First, the follow-up period after the treatment was limited, for the majority of the treated it was two years at maximum. These results merit confirmation using a longer follow-up period. Second, the employees and the wards in the treatment and control group may differ by many unmeasured confounders. We did control for the unmeasured individual-level confounders, as mentioned before. We also included the not-yet-treated individuals in the control group. We also controlled for potential ward-level correlation by using ward-level clustered errors. But we were unable to establish full causality between using participatory scheduling tools and sickness absence, as there could be plenty of time-varying factors that we could not control in our models. Third, the results may have limited generalizability outside the shift work carried out in the healthcare sector. The results are based on the Finnish healthcare sector which utilizes rather short, 3-week shift scheduling periods. Fourth, we have no information on the actual implementation process of the shift scheduling tools. We took a conservative measure of the participatory scheduling intervention by having an 80% threshold as a measure of the intervention. Fifth, there is selection into the double-intervention group since the use of the shift evaluation tool was, to the best of our knowledge, voluntary. The results of the use of the shift evaluation tool should thus be interpreted with caution. However, we offer an indication that it is important to have a view of the ergonomics of shift work schedules while self-rostering.

The results of this study are important to not only hospital management responsible for shift scheduling but also to all workplaces that plan to implement self-rostering. The effects of irregular working hours may be potentially negative for health and well-being. The effects of sickness absence were increasing, yet delayed i.e., took a rather long time to emerge but can be improved by combining self-rostering with the automatic evaluation of shift ergonomics giving feedback to the employees on the possible risks of the desired individual working hour patterns. The systematic evaluation and feedback to the employees on the possible health risks of the working hours should be considered when self-rostering is implemented. The potential to lower labor costs by self-rostering may need a targeted focus on working hour characteristics.

## 5. Conclusions

Among the studied hospital employees, the use of the participatory scheduling tool introduced an increase of compressed working hours and longer shifts, and a delayed, increasing effect of approximately 20% on the short sickness absence days and all sickness absence episodes. The use of the shift schedule evaluation tool in combination with the participatory shift scheduling tool attenuated the observed effects on working hour characteristics and sickness absence. For self-rostering of irregular working hours in shift work, it might be beneficial to focus on the evaluation and feedback of shift ergonomics.

## Figures and Tables

**Figure 1 ijerph-19-14654-f001:**
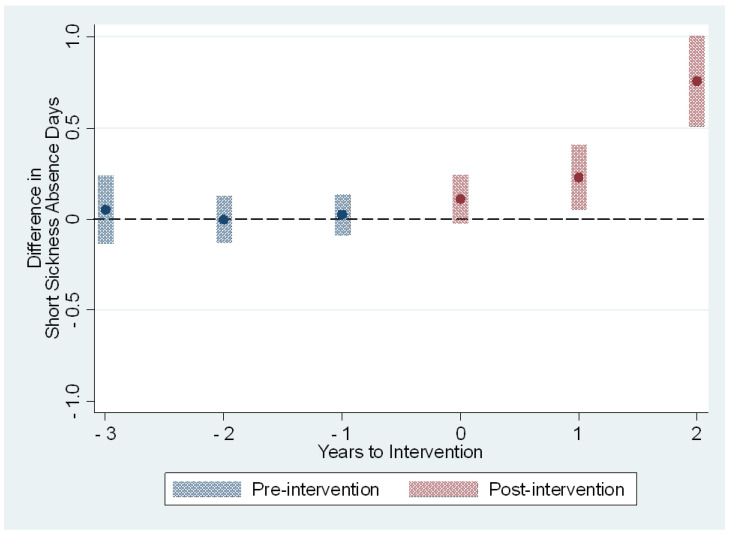
The estimated dynamic average treatment effects of the use of the participatory scheduling tool on sickness absence days from short absences of 1–3 days. Event study estimates with 95% confidence intervals depict difference in outcome measure between intervention and control groups relative to timing of the intervention. Year 0 is the first full year of the intervention.

**Figure 2 ijerph-19-14654-f002:**
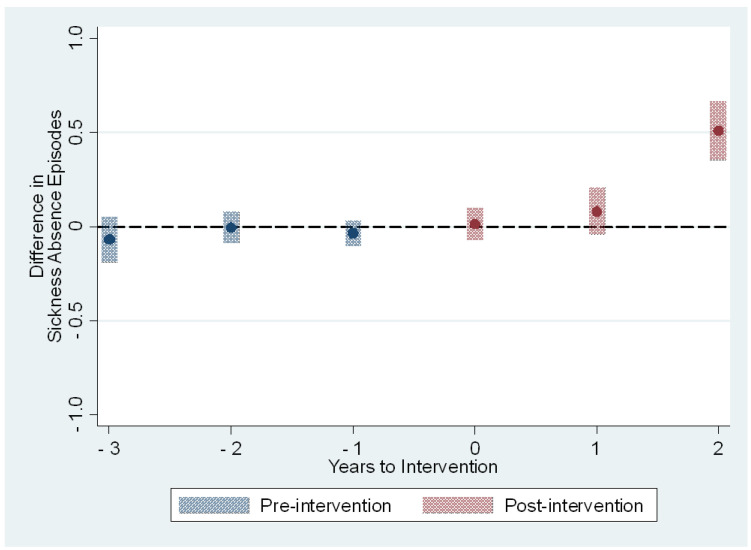
The estimated dynamic average treatment effects of the use of the participatory scheduling tool on sickness absence episodes from all sickness absences. Event study estimates with 95% confidence intervals depict difference in outcome measure between intervention and control groups relative to timing of the intervention. Year 0 is the first full year of the intervention.

**Figure 3 ijerph-19-14654-f003:**
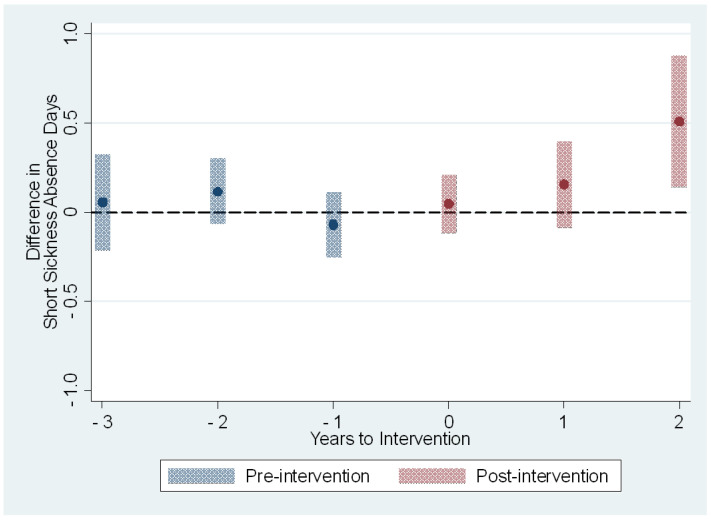
The estimated dynamic average treatment effects of the use of the participatory scheduling tool with shift schedule evaluation tool on sickness absence days from short sickness absences of 1–3 days. Event study estimates with 95% confidence intervals depict difference in outcome measure between intervention and control groups relative to timing of the intervention. Year 0 is the first full year of the intervention.

**Figure 4 ijerph-19-14654-f004:**
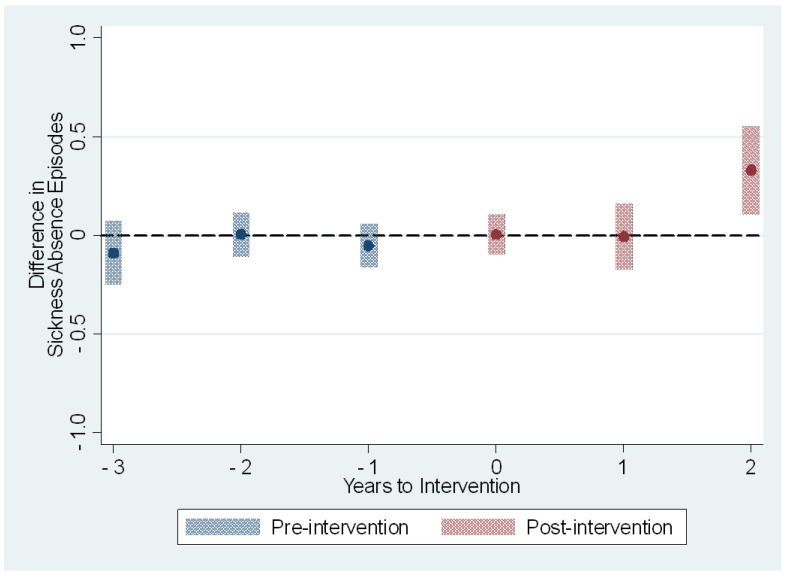
The estimated dynamic average treatment effects of the use of the participatory scheduling tool with shift schedule evaluation tool on sickness absence episodes from all sickness absences. Event study estimates with 95% confidence intervals depict difference in outcome measure between intervention and control groups relative to timing of the intervention. Year 0 is the first full year of the intervention.

**Table 1 ijerph-19-14654-t001:** Baseline characteristics (2015) of the intervention and never-treated control groups. *n* = 16,557. Left panel, the use of participatory scheduling tool, *n* = 6143 and 10,345. Right panel, the use of participatory scheduling tool with the shift schedule evaluation tool, *n* = 2014 and 14,474. Note that *, **, *** denote statistical significance at 0.05, 0.01, and 0.001 levels, respectively.

	The Primary Intervention, i.e., the Use of the Participatory Scheduling Tool			The Secondary Intervention, i.e., the Use of the Participatory Scheduling Tool with Shift Schedule Evaluation Tool		
	Intervention, *n* = 6143	Control, *n* = 10,345	Difference between Intervention and Control	Double-Intervention, *n* = 2014	Control, *n* = 14,474	Difference between Intervention and Control
	Mean (SD)	Mean (SD)		Mean (SD)	Mean (SD)	
Women, % Nursing profession, %	88.3 84.6	84.4 63.6	3.9 21.0 ***	89.1 85.5	85.4 69.5	3.7 16.0 ***
Age, Years	41.9 (10.6)	42.9 (10.7)	−0.9 ***	39.7 (10.6)	42.9 (10.6)	−3.2 ***
Number of employees in ward	29.6 (18.0)	27.3 (19.4)	2.3 ***	29.2 (16.0)	28.0 (19.3)	1.2 **
Share of evening shifts, %	32.3 (14.2)	32.3 (15.9)	0.0	32.1 (13.6)	32.3 (15.5)	−0.2
Share of night shifts, %	20.1 (20.1)	16.4 (20.4)	3.7 ***	19.3 (17.8)	17.6 (20.7)	1.7 ***
>40-h working weeks, %	28.2 (13.4)	28.2 (14.5)	0.0	28.8 (12.8)	28.1 (14.5)	0.7 *
>48-h working weeks, %	6.5 (7.3)	7.0 (8.1)	−0.4 ***	6.5 (7.1)	6.8 (7.9)	−0.3
>12-h shifts, %	6.1 (10.2)	7.4 (13.6)	−1.3 ***	5.9 (10.2)	7.0 (12.7)	−1.1 ***
Long spells of work shifts, %	3.8 (5.8)	4.1 (6.7)	−0.3 **	3.9 (5.2)	4.0 (6.5)	−0.1
<11-h shift intervals, %	18.3 (11.6)	15.0 (12.5)	3.3 ***	18.3 (11.2)	15.9 (12.4)	−2.4 ***
<28-h recovery periods after last night shift, %	4.7 (13.6)	5.5 (15.4)	−0.8 ***	4.8 (13.7)	5.3 (14.9)	−0.5
Realized shift plans, %	91.5 (7.1)	91.1 (7.9)	0.3 **	91.5 (6.9)	91.2 (7.7)	0.3
Weekend work, %	41.9 (17.0)	35.6 (19.6)	6.2 ***	41.8 (16.2)	37.4 (19.2)	4.3 ***
Single free days, %	14.1 (7.8)	13.7 (8.8)	0.3 **	14.7 (7.7)	13.8 (8.5)	1.0 ***
Days from short sickness absences of 1–3 days	3.6 (3.8)	3.5 (3.9)	0.1	3.4 (3.5)	3.5 (3.9)	−0.2
Episodes from short sickness absences of 1–3 days	2.0 (2.0)	2.0 (2.1)	0.0	1.9 (1.9)	2.0 (2.1)	−0.1 *
Days from all sickness absences	12.4 (21.8)	12.0 (21.0)	0.4	11.4 (19.9)	12.3 (21.5)	−0.9
Episodes from all sickness absences	2.7 (2.6)	2.7 (2.6)	0.0	2.5 (2.4)	2.7 (2.6)	−0.1 *

**Table 2 ijerph-19-14654-t002:** The estimated effects of the use of the participatory scheduling tool and the shift schedule evaluation tool on the occurrence of various shift work characteristics. Event study estimates with ward-level clustered standard errors. Statistically significant differences are bolded.

	The Difference in Outcome Measures					
	Before the Intervention			After the Intervention		
	Year 3	Year 2	Year 1	Year 1	Year 2	Year 3
	Percentage Points (95% CI)	Percentage Points (95% CI)	Percentage Points (95% CI)	Percentage Points (95% CI)	Percentage Points (95% CI)	Percentage Points (95% CI)
(i) Participatory scheduling vs. Never-treated and Not-yet-treated						
>40-h working weeks, %	0.3 (−0.4; 1.1)	−**0.5** **(−1.0; −0.0)**	0.1 (−0.3; 0.4)	**0.9** **(0.5; 1.3)**	**0.6** **(0.1; 1.2)**	0.8 (−0.1; 1.6)
>48-h working weeks, %	−0.2 (−0.7; 0.3)	−0.3 (−0.7; 0.2)	**0.3** **(0.0 0.5)**	**0.6** **(0.4; 0.8)**	**0.5** **(0.2; 0.9)**	**0.8** **(0.3; 1.3)**
>12-h shifts, %	0.0 (−0.4; 0.4)	0.0 (−0.2; 0.3)	0.2 (−0.0; 0.4)	**0.4** **(0.1; 0.6)**	0.4 (−0.1; 0.8)	**0.8** **(0.1; 1.5)**
Long spells of work shifts, %	0.1 (−0.3; 0.5)	−0.1 (−0.5; 0.2)	**0.3** **(0.2; 0.5)**	0.2 (−0.0; 0.3)	0.1 (−0.2; 0.3)	0.2 (−0.2; 0.5)
<11-h shift intervals, %	0.4 (−0.1; 0.8)	0.2 (−0.1; 0.5)	−0.1 (−0.4; 0.2)	0.3 (0.0; 0.5)	0.4 (−0.1; 0.9)	0.6 (−0.0; 1.3)
<28-h recovery periods after last night shift, %	−0.1 (−0.8; 0.7)	−0.6 (−1.3; 0.0)	0.0 (−0.4; 0.5)	0.5 (0.0; 1.0)	0.3 (−0.4; 1.1)	**1.0** **(0.1; 1.9)**
Realized shift plans, %	−0.2 (−0.5; 0.1)	−0.1 (−0.3; 0.2)	**0.2** **(0.0; 0.4)**	0.1 (−0.1; 0.4)	−0.1 (−0.4; 0.2)	0.0 (−0.4; 0.4)
Weekend work, %	**1.00** **(0.3; 1.7)**	0.0 (−0.5; 0.5)	**0.9** **(0.6; 1.3)**	**1.8** **(1.5; 2.2)**	**2.6** **(2.0; 3.1)**	**4.0** **(3.1; 4.9)**
Single free days, %	0.2 (−0.2; 0.6)	0.2 (−0.0; 0.5)	**0.3** **(0.0; 0.5)**	**0.4** **(0.2; 0.6)**	**0.7** **(0.3; 1.0)**	**0.8** **(0.3; 1.3)**
(ii) Participatory scheduling and shift evaluation vs. Never-treated and Not-yet-treated						
>40-h working weeks, %	0.7 (−0.2; 1.7)	−0.5 (−1.2; 0.3)	0.4 (−0.1; 0.9)	**0.9** **(0.4; 1.4)**	0.6 (−0.1; 1.4)	0.5 (−0.8; 1.7)
>48-h working weeks, %	0.0 (−0.5; 0.6)	−0.3 (−0.8; 0.3)	0.3 (−0.1; 0.6)	**0.5** **(0.2; 0.8)**	0.2 (−0.3; 0.6)	**0.9** **(0.1; 1.6)**
>12-h shifts, %	−0.1 (−0.5; 0.2)	0.0 (−0.2; 0.3)	0.1 (−0.1; 0.4)	0.2 (−0.1; 0.4)	0.4 (−0.1; 0.8)	0.9 (0.0; 1.8)
Long spells of work shifts, %	0.0 (−0.4; 0.5)	−0.1 (−0.5; 0.4)	**0.2** **(0.0; 0.4)**	0.1 (−0.2; 0.3)	−0.2 (−0.6; 0.2)	−0.4 (−0.9; 0.1)
<11-h shift intervals, %	0.0 (−0.6; 0.7)	**0.5** **(0.1; 0.9)**	0.0 (−0.3; 0.4)	0.3 (−0.1; 0.6)	0.3 (−0.3; 0.9)	−0.0 (−0.9; 0.9)
<28-h recovery periods after last night shift, %	0.0 (−1.4; 1.3)	−0.7 (−1.6; 0.3)	0.4 (−0.4; 1.1)	−0.1 (−0.8; 0.6)	−0.7 (−1.7; 0.4)	1.4 (−0.3; 3.1)
Realized shift plans, %	0.1 (−0.3; 0.5)	−0.1 (−0.5; 0.2)	0.2 (−0.1; 0.4)	0.2 (−0.1; 0.4)	−0.2 (−0.6; 0.3)	−0.1 (−0.8; 0.6)
Weekend work, %	**1.3** **(0.4; 2.1)**	**0.9** **(0.3; 1.5)**	**1.1** **(0.6; 1.6)**	**1.3** **(0.8; 1.8)**	**0.8** **(0.0; 1.6)**	0.7 (−0.6; 1.9)
Single free days, %	0.3 (−0.2; 0.8)	**0.7** **(0.3; 1.0)**	0.3 (−0.1; 0.6)	**0.4** **(0.1; 0.7)**	0.2 (−0.4; 0.8)	0.7 (−0.2; 1.7)

(i)—Specification estimates the effects of the use of participatory scheduling tool; (ii)—Specification estimates the effects of the use of participatory scheduling with the shift schedule evaluation tool; —Specifications control for the age, sex, nursing profession, ward size and the share of evening and night shifts out of all shifts. The estimated effect is the average treatment effect on the treated measured as percentage points of the occurrence of shift work characteristic.

**Table 3 ijerph-19-14654-t003:** The estimated effects of the use of the participatory scheduling tool and the shift schedule evaluation tool on sickness absence days from short absences of 1–3 days and episodes from all absences. Event study estimates with ward-level clustered standard errors. Statistically significant differences are bolded.

	The Difference in Outcome Measures					
	Before the Intervention			After the Intervention		
	Year 3	Year 2	Year 1	Year 1	Year 2	Year 3
	Days (95% CI)	Days (95% CI)	Days (95% CI)	Days (95% CI)	Days (95% CI)	Days (95% CI)
(i) Participatory scheduling vs. Never-treated and Not-yet-treated						
Days from short sickness absences of 1–3 days	0.1 (−0.1; 0.2)	0.0 (−0.1; 0.1)	0.0 (−0.1; 0.1)	0.1 (0.0; 0.2)	**0.2** **(0.0; 0.4)**	**0.8** **(0.5; 1.0)**
(ii) Participatory scheduling and shift evaluation vs. Never-treated and Not-yet-treated						
Days from short sickness absences of 1–3 days	0.1 (−0.2; 0.3)	0.1 (−0.1; 0.3)	−0.1 (−0.2; 0.1)	0.0 (−0.1; 0.2)	0.2 (−0.1; 0.4)	**0.5** **(0.1; 0.9)**
	Before the intervention			After the intervention		
	Year 3	Year 2	Year 1	Year 1	Year 2	Year 3
	Episodes (95% CI)	Episodes (95% CI)	Episodes (95% CI)	Episodes (95% CI)	Episodes (95% CI)	Episodes (95% CI)
(i) Participatory scheduling vs. Never-treated and Not-yet-treated						
Episodes from all sickness absences	−0.1 (−0.2; 0.1)	0.0 (−0.1; 0.1)	0.0 (−0.1; 0.0)	0.0 (−0.1; 0.2)	0.1 (−0.1; 0.2)	**0.5** **(0.4; 0.7)**
(ii) Participatory scheduling and shift evaluation vs. Never-treated and Not-yet-treated						
Episodes from all sickness absences	−0.1 (−0.3; 0.1)	0.0 (−0.1; 0.1)	−0.1 (−0.2; 0.1)	0.0 (−0.1; 0.1)	0.00 (−0.2; 0.2)	**0.3** **(0.1; 0.6)**

(i)—Specification estimates the effects of the use of participatory scheduling tool; (ii)—Specification estimates the effects of the use of participatory scheduling and the shift schedule evaluation tool; —Specifications control for the age, sex, nursing profession, ward size, the proportion of evening and night shifts out of all shifts, and shift work characteristics including the percentages of >40-h working weeks, >48-h working weeks, >12-h shifts, long spells of work shifts, <11-h shift intervals, <28-h recovery periods after last night shift, realized shift plans, weekend work, and single free days. The estimated effect is the average treatment effect on the treated measured as days or episodes of sickness absence.

## Data Availability

The research data are not publicly available due to privacy and ethical restrictions. Research data are not available on request from the authors or FIOH due to written agreements with the register data owners (the hospital districts) not to forward any data to third parties.

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
