# Peer review of "The Time-Varying Effect of Participatory Shift Scheduling on Working Hour Characteristics and Sickness Absence: Evidence from a Quasi-Experiment in Hospitals"

_ijerph, 2022, doi:10.3390/ijerph192214654_

Round 1

Reviewer 1 Report

The authors presented interesting analyses of the results of the impact of the work organisation of hospital employees on sickness absence. They showed that a participatory shift work scheduling system contributes to it. In addition, a tool to assess the level of ergonomics of shift scheduling was used and a decrease in the absence rate was observed. The authors reported that they were aware of the limitations of their study.

Specific comments:

1) Chapter 1 introduces a number of theses without citing the source literature (lines 100 to 123);

2) Chapter 2.1 characterises the study population in a very general way. The occupational groups in the hospital departments on which the study was made, were not specified in detail. Conditions differing in work hardness and exposures have an impact on the absence rate.

3) Section 2.2: What was the purpose of selecting short-term absences (1-3 days)?

Was the impact of working conditions and biomechanical loads of employees on staff sickness absence analysed? Were direct and indirect causes of the absences identified and analysed?

4) I think it is worth mentioning more about the shift schedule assessment tool. It is written that it has been used, but the criteria used in it to assess the level of ergonomics are not indicated.

5) Lines 601-604: I think it is worth to mention the unmeasured factors that were not taken into account.

Author Response

Response to comments made by Reviewer 1:

POINT 1: Chapter 1 introduces a number of theses without citing the source literature (lines 100 to 123);

Our response: Thank you for all your comments and suggestions. We have now improved the citation used in rows 100-110.

Rows 100-110 were updated as follows “The studies [28-33] on self-rostering in shift work offer somewhat mixed evidence. The previous findings indicate that while self-rostering and participatory shift scheduling improve WTC, health and well-being, irregularity of working hours may increase as a result. The findings also suggest that the increase in irregularity of working hours (e.g., variability of shift starting and ending times, and the length of the work shifts and spells), may have adverse health effects and consequently lead to increased sickness absence [10-11, 14-17]. On the other hand, the use of shift schedule evaluation tool may improve the ergonomics of shift schedules and thus offer ways to reduce those working hour characteristics in irregular shift work that may have negative effects on health and wellbeing (e.g., long work shifts and quick returns, i.e., evening-morning shift combinations) [20].”

Following response to reviewer 2 point 3, rows 111-119 now read “The focus of this study is the investigation of the effects of self-rostering to shifts by using the participatory shift scheduling tool alone or in combination with the shift schedule evaluation tool. The participatory scheduling tool offers an opportunity to individual employee to self-roster in to shifts [see e.g., 32]. The shift schedule evaluation tool offers feedback in evaluating the ergonomics of shift schedules [see e.g., 20]. Together the two shift scheduling tools, the participatory shift scheduling tool and the shift schedule evaluation tool studied here can have both impairing and improving effects on working hour characteristics, irregular hours in shift work, and consequently, both increasing and decreasing effects on sickness absence.”

POINT 2: Chapter 2.1 characterises the study population in a very general way. The occupational groups in the hospital departments on which the study was made, were not specified in detail. Conditions differing in work hardness and exposures have an impact on the absence rate.

Our response: Thank you for this very important note. We have added information on study population / sample accordingly to Table 1 and Results section. As there was a difference of 21 percentage points in the share of nursing personnel between the intervention and control groups, we have re-run our analysis. In more detail, we controlled for the employees nursing profession in all our regression analyses.

Manuscript has been updated accordingly. The changes to the manuscript were made using track changes. There were no changes in the sickness absence results nor in the conclusions of the study.

POINT 3: Section 2.2: What was the purpose of selecting short-term absences (1-3 days)?

Was the impact of working conditions and biomechanical loads of employees on staff sickness absence analysed? Were direct and indirect causes of the absences identified and analysed?

Our response: We did not analyse the impact of working conditions (other than working hours), biomechanical loads since they were not available in our data, or direct or indirect causes as it was not the focus of this study.

We have added the purpose of selecting the short-term absences in the introduction accordingly.

Rows 123-130 now read “Our focus was on short-term sickness absence days, but we also calculated the total number of sickness absence episodes irrespective of the length of the episode. Short-term sickness absence was selected as the focus because they can potentially reflect self-perceived health or motivation to work, i.e., they are not always associated with disease condition or symptoms of any kind [5]. However, we also included sick-ness absence episodes irrespective of their length, as the increase of short-sickness absences can potentially be observed as changes in the total number of sickness absence episodes. “

We also added a sentence on diagnosis information on rows 164-166 in section 2.2. as follows. “We had no information on diagnosis-specific or medically certified sickness absence.”

POINT 4: I think it is worth mentioning more about the shift schedule assessment tool. It is written that it has been used, but the criteria used in it to assess the level of ergonomics are not indicated.

Our response: We agree with this comment and have supplemented the text in the Methods section (rows 195-204) as follows: “The shift schedule evaluation tool offers feedback and help in evaluating the work schedules according to timing and length of working hours, number of consecutive night shifts and recovery time between shifts. In the standard version of the software, the evaluation tool can be used only by the shift planner of the ward. In the participatory shift scheduling tool, the shift schedule evaluation tool can be used by both the shift planner and the employees. [20] Basically, the schedule evaluation tool can be used to detect the potential health and well-being associated drawbacks in the working hour schedule, and to revise the schedule if desired. As opposed to the use of the participatory scheduling tool, the use of the shift schedule evaluation tool by individual employees was though voluntary. “

POINT 5: Lines 601-604: I think it is worth to mention the unmeasured factors that were not taken into account.

Our response: Thank you for this important note. We control for unmeasured individual-level factors that do not change in time. We also control for time-varying factors that are common to all individuals in the sample. We do not control for unmeasured time-varying individual-level factors. While it is impossible to include a list of all unmeasured time-varying factors to the manuscript, we have added a following sentence in rows 604-607. “However, the unmeasured time-varying individual-level factors, such as life-style factors, cannot be statistically controlled in our regression analyses.“

Please see the attachment for our reply to Reviewer 2 and the Editors.

Reviewer 2 Report

I appreciate the hard work invested in this study/manuscript.

I will present only the few aspects that need to be addressed by the authors:

-         = The title includes periods. Punctuation in the actual title should be changed

-         =  In the Abstract section but also in the body of the manuscript the brackets for confidence intervals used by the authors are (). They should be [].

-          = The Introduction section does not include definitions/presentation of what is a shift schedule evaluation tool, participatory shift scheduling tool and self-rostering shift scheduling. It would worth to provide a definition and a clear distinction between these shift scheduling types.

-          = The Methods section includes information about 2.1. Population. It should be appropriate to use the term Sample of Participants

-        =  Table 1 include in the first row (Statistical significance). It is redundant as the statistical significance is indicated by the use of the conventional signs, such as ***, **

-         = Some notions such as “treatment”, “pre-treatment”, “post-treatment” can be replaced with “intervention”, “pre-intervention”, “post-intervention”, group in the condition XX

Author Response

Response to comments made by Reviewer 2:

POINT 1: The title includes periods. Punctuation in the actual title should be changed

Our response: Thank you for all your comments and suggestions. We have updated the title accordingly. The updated title is “The time-varying effect of participatory shift scheduling on working hour characteristics and sickness absence: Evidence from a quasi-experiment in hospitals”.

POINT 2: In the Abstract section but also in the body of the manuscript the brackets for confidence intervals used by the authors are (). They should be [].

Our response: We have updated the brackets used for confidence intervals in the Abstract and elsewhere in the manuscript as suggested. We have not updated the brackets for confidence intervals in tables 2 and 3 or appendix table 1.

POINT 3: The Introduction section does not include definitions/presentation of what is a shift schedule evaluation tool, participatory shift scheduling tool and self-rostering shift scheduling. It would worth to provide a definition and a clear distinction between these shift scheduling types.

Our response: We have now improved the introduction as follows (rows 111-119): “The focus of this study is the investigation of the effects of self-rostering to shifts by using the participatory shift scheduling tool alone or in combination with the shift schedule evaluation tool. The participatory scheduling tool offers an opportunity to individual employee to self-roster in to shifts [see e.g., 32]. The shift schedule evaluation tool offers feedback in evaluating the ergonomics of shift schedules [see e.g., 20]. Together the two shift scheduling tools, the participatory shift scheduling tool and the shift schedule evaluation tool studied here can have both impairing and improving effects on working hour characteristics, irregular hours in shift work, and consequently, both increasing and decreasing effects on sickness absence.”

Please see also Our response 4 to Reviewer 1.

POINT 4: The Methods section includes information about 2.1. Population. It should be appropriate to use the term Sample of Participants

Our response: We have updated the term used as suggested. 

POINT 5: Table 1 include in the first row (Statistical significance). It is redundant as the statistical significance is indicated by the use of the conventional signs, such as ***, **

Our response: We have removed the term “(Statistical significance)” as suggested.

POINT 6: Some notions such as “treatment”, “pre-treatment”, “post-treatment” can be replaced with “intervention”, “pre-intervention”, “post-intervention”, group in the condition XX

Our response: We have updated the “treatment”-terms to “intervention”-terms throughout the manuscript including graphs and tables. This introduced plenty of changes that can be viewed in the edited manuscript. The term “treatment” and related “treated” were left unchanged when they were included as a part in the recognised methodological jargon, for example in, “average treatment effect on the treated”.

Please see the attachment for our reply to Reviewer 1 and the editors.
